# Lower Limb Muscle Strength Matters: Effect of Relative Isometric Strength on Countermovement and Rebound Jump Performance in Elite Youth Female Soccer Players

**DOI:** 10.3390/muscles4030023

**Published:** 2025-07-10

**Authors:** Jack Fahey, Paul Comfort, Nicholas Joel Ripley

**Affiliations:** 1Directorate of Psychology and Sport, School of Health and Society, University of Salford, Salford M6 6PU, UK; p.comfort@salford.ac.uk (P.C.); n.j.ripley@salford.ac.uk (N.J.R.); 2Department of Football Medicine and Science, Manchester United Football Club, Carrington Training Complex, Manchester M31 4AB, UK; 3Strength and Power Research Group, School of Medical and Health Sciences, Edith Cowan University, Joondalup, WA 6027, Australia

**Keywords:** girls’ football, plyometric, maximal force, training, hip extensors, knee extensors, plantar flexors

## Abstract

Background: Expression of maximal and rapid force in the hip, knee, and plantar flexors is important for athletic performance in female soccer. This study was designed to determine the effect of relative isometric strength in the isometric mid-thigh pull (IMTP) on countermovement jump (CMJ) and rebound jump (CMJ-R) performance in female youth soccer players. Methods: Ninety-six female soccer players (age: 14.1 ± 2.3 years, height: 160.5 ± 9.7 cm, mass: 55.0 ± 10.3 kg) completed three trials of the IMTP, CMJ, and CMJ-R using force plates. Players were categorized as stronger (top quartile, *n* = 19) and weaker players (bottom quartile, *n* = 23) based on IMTP relative net peak force. One-way Bayesian independent *t*-tests were performed between stronger and weaker players. Results: Very large difference in lower limb strength between groups (*g* = 5.05). Moderate to very strong evidence to support that stronger players had greater countermovement depth and time to take-off with similar jump heights than weaker players. Strong evidence was observed for CMJ-R height, with stronger players falling from a greater height and executing similar ground contact times compared to weaker players. Conclusions: Relative strength has implications for CMJ-R, highlighting the importance of developing relative strength in hip, knee, and plantar extensors.

## 1. Introduction

Female football has grown exponentially in the past decade owing to the implementation of specific female continental and global strategies to develop participation, provisions, and performance from grassroots to elite and international levels [1,2,3,4,5,6]. The professionalization of female soccer and increase in sport science and medical provisions have led to an increase in the physical match demands with increasing age and competition level [7,8]. Research in female football has also increased, mainly focusing on strength and conditioning [9,10], and injury risk mitigation [11,12]. Researchers have provided recommendations for future research suggesting further descriptive studies followed by the physiological and contextual influences on female health and performance [13,14], sex-matched research designs [15], a future critical expert evaluation of the existing quantity and identification of appropriate themes [9] and interventions to support female health and performance [9,13,14,15]. Previous researchers have also documented the importance of high-intensity movements, including jumping, sprinting, and change in direction, for athletic and sporting success in female football [16]. Despite this, the quantity of research in women’s football is much lower than research in men’s football [9], while other researchers have reported a lower frequency of strength and conditioning sessions in females compared to males [10].

Non-bone-related injury risk (i.e., ligament and/or soft tissue injuries) is greater in female football players compared to males, with greater injury incidence in training anterior cruciate ligament injury (ACL) [12,17,18,19,20,21]. The most common injury location in female footballers is the knee [11], with greater incidence occurring during matches compared to training (19.2 per 1000 h of exposure vs. 3.5 per 1000 h of exposure, respectively) [18,19]. In female youth footballers, Robles et al. [22] reported that similar trends are observed with injury incidence compared to male youth footballers (6.77 vs. 5.70 per 1000 h of exposure, respectively). Previous injury is a well-established non-modifiable risk factor for future injury; however, this may not be applicable to young footballers who have yet to experience non-bone-related injury. Instead, practitioners would be more impactful designing interventions to reduce modifiable risk factors such as relative strength [23,24]. However, previous researchers have demonstrated no change in relative strength with age, despite increases in body mass and sprinting and change in direction speeds (i.e., greater momentum) [25], which presents a problem for youth female footballers aiming to transition into senior age groups and higher levels of competition, or simply biological differences that occur during peak height velocity. Maximal muscular strength is associated with superior rapid force production [26,27], athletic performance (i.e., jumping and sprinting) [28], decreased injury risk [23,29], and is crucial for goal scoring opportunities during matches [16]. High levels of maximal and rapid force production are important as net force relative to mass determines acceleration, with the duration of force application (i.e., net impulse [∆force × ∆time] relative to the mass being accelerated) determining the resulting velocity. As football incurs time pressure situations (i.e., goal scoring opportunities, tackles and changes in possession), it is important for youth female footballers to increase their relative strength (and subsequently rapid force production), which are likely positive implications on injury risk and performance.

Force plates enable high resolution assessment of force–time characteristics relating to Newton’s third law (i.e., every action has an equal and opposite reaction) [30]. Portable force plates are now accessible and used by 50% of strength and conditioning coaches in football [31]. Force plates can be used across a range of dynamics and isometric tasks to provide valid [32] and reliable assessment of force–time characteristics [33], providing a proxy for neuromuscular function (including neuromuscular fatigue). The most common test used in team sports is the bilateral countermovement jump (CMJ) [34], with the most frequently reported metrics being jump height, peak force, and peak power [34]. Harper et al. [35] assessed the CMJ using force plates and was able to differentiate between football players of high and low deceleration ability based on peak braking and propulsive force and velocity. Despite the utility of the CMJ, this reflects ballistic performance, whereas a measure of reactive (or plyometric) ability is required to better understand athletic performance. The countermovement rebound jump (CMJ-R) is a reliable [33], alternative test to the drop jump, which is the most common test used to assess fast stretch shortening cycle performance [36,37]. Researchers have demonstrated different fall heights with the drop jump test [38], whereas the CMJ-R enables participants to fall from their maximum CMJ height, creating a standardized relative fall height.

Practitioners can also use force plates for isometric tasks, including the isometric mid-thigh pull (IMTP), with clear standardization and methodological recommendations provided by Comfort et al. [39]. Soriano et al. [40] reported that senior and national footballers were stronger than young and regional players, highlighting the importance of ‘*being strong*’. Further, peak and rapid force production during the IMTP is associated with sprint and change in direction speed [41]. The IMTP has been used to monitor lower limb maximal strength in female youth footballers [25,33,42,43,44]. Emmonds et al. [25] reported no change in IMTP relative peak force with increasing age despite increases in body mass and sprinting speed (and therefore momentum), potentially highlighting limitations within their strength training regimes. In contrast, Fahey et al. [33] reported moderate increases in relative net peak force following six weeks of combined strength and plyometric training in elite female youth footballers, highlighting its importance and potential to improve relative strength.

Krzyszkowski et al. [45] assessed phase-specific predictors of CMJ performance in good vs. bad jumpers, based on modified reactive strength index (mRSI), using conventional statistical approaches (i.e., null hypothesis testing and effect sies). Bayesian statistics have been proposed as a more robust method to make inferences in sport science, in particular where low sample sizes are often observed [46]. Bayesian statistics also provide a better probabilistic approach when determining differences between two defined groups and can offer simplistic statements for interpretation [47]. This approach considers parameters that have a true but unknown value described by a posterior probability, reflecting uncertainty [47]. This is calculated as follows:Likelihood × prior

Likelihood is the probability of observing the data, whilst the prior is the probability of obtaining these independently of the data. Priors can be determined from previous experiments and/or expert opinions. Priors can also be uninformative to allow for inferences to be drawn by using a default method [47]. Ripley et al. [48] used Bayesian statistics to determine the influence of relative strength on CMJ performance in professional and semi-professional male footballers and observed weak evidence to support that stronger players had greater jump height, however they did display supe-rior temporal metrics (i.e., reduced time to take-off) (i.e., IMTP gross relative peak force > 33.1 N/kg). Considering the rapid growth of female football, yet increased injury incidence and risk compared to males, maximal and rapid force expression are likely to have positive implications on injury risk and performance. As net force relative to body mass determines acceleration, which is an important physical attribute in female football [16], information pertaining to the influence of relative strength on CMJ and CMJ-R performance in female youth footballers would enable practitioners to design strength and conditioning programs and interventions to improve female health and performance [9,13,14,15]. Therefore, the purpose of this study was to determine the effect of relative strength on CMJ and CMJ-R performance in elite female youth footballers and to offer practitioners valuable insights on the importance of relative strength on CMJ and CMJ-R performance. It was hypothesized that ‘*stronger*’ players would achieve superior CMJ and CMJ-R performance than ‘*weaker*’ players relating to outcome (i.e., greater jump height), driver (i.e., greater relative average force), and strategy (i.e., shorter time to take-off).

## 2. Materials and Methods

### 2.1. Experimental Setup

An observational cross-sectional design was used to determine the effect of isometric strength relative to body mass on CMJ and CMJ-R performance in elite female youth footballers. Testing was performed mid-season (November 2024), when the players had accumulated training and competitive match play volume. This testing period was chosen rather than pre-season (which may reflect a detrained state) or end of season (which may involve fatigue). All tests were conducted a minimum of 48 h following a competitive match, with no intense training taking place during this period. Each player completed a minimum of three and a maximum of five repetitions on the CMJ, CMJ-R, and IMTP, which were completed in a random order to reduce order effects. For players over the age of 18 years, written informed consent was provided, whereas for players under the age of 18 years, written parental/guardian ascent was provided. Institutional ethics approval was granted (ID = 13428).

### 2.2. Participants

Ninety-six female youth footballers (age: 14.1 ± 2.3 years, height: 160.5 ± 9.7 cm, mass: 55.0 ± 10.3 kg) registered with a Tier 1 Plus Football Association Academy volunteered for the study. Players were considered highly trained to elite according to McKay et al. [49], participating in three technical training sessions, one to three strength and conditioning sessions, and one competitive match per week. Players trained and played matches in chronological age groups (U11, U12, U13, U14, U15, U16, and U21). All players were free from injury prior to testing.

### 2.3. Procedures

On arrival at the training facility, participants’ stature was measured (Seca, Birmingham, UK) to the nearest 0.1 cm before a standardized RAMP warm-up consisting of dynamic stretches and movements including squats, lunges, hops, and submaximal jumps. Players were familiar with the testing protocols, having completed these as part of routine testing. All tests were conducted on valid [32,50] and reliable [33] dual sensor force plates (Hawkin Dynamics Inc., Westbrook, ME, USA), sampling at 1000 Hz and proprietary software. Foam pads were positioned to surround the force plates during the CMJ and CMJ-R for player safety, whereas the IMTP was conducted on a metal customized rig (Absolute Strength, Cardiff, UK). For the CMJ and CMJ-R, players were required to stand centrally on the force plates with arms akimbo for a one-second weighing period before executing each jump. Players were allowed to self-select the countermoveing depth, with verbal instructions provided to each player to “jump as high and fast as possible”. These instructions relate to specific phases of the CMJ and CMJ-R, for example, “jump as fast” refers to performing the countermovement, propulsive, and rebound phases as quickly as possible. Additional trials were performed if participants removed their hands from their hips during the CMJ or CMJ-R, executed a slower countermovement, tucked their knees during the flight phases of both jumps, and a reduction >10% in CMJ portion of the CMJ-R was observed. The IMTP was conducted as recommended by Comfort et al. [39], ensuring that the posture reflected the start of the second pull phase of a clean; 125–145° knee angle and 140–150° hip angle, and with weightlifting straps to eliminate grip strength as a limiting factor. Players were required to remain as still as possible for at least 1 s to allow calculation of system mass before executing the task with consistent instruction to as ‘push as hard and as fast as possible for 3–5 s’ with strong verbal encouragement to ‘push’ provided throughout. Additional trials were included in a difference of >250 N was achieved between tests and a loss of technique.

### 2.4. Data Analyses

Vertical ground reaction force was low-pass filtered at 50 Hz [51]. For the CMJ and CMJ-R, all metrics were calculated automatically by the force plate proprietary software as recommended by McMahon et al. [52]. The onset for identifying the start of the jump (i.e., unweighting phase) was identified from the point when the vertical ground reaction force exceeded five standard deviations (SDs) of the average force during the one-second weighing period and continued until the negative vertical force returned to within the five SD threshold. The start of the braking phase was identified at the end of the unweighting phase, from peak negative velocity of the center of mass until vertical velocity reached zero. The propulsive phase was identified from the end of the braking phase, when force returned to body mass, and continued until vertical force dropped below 25 N. Time to take-off was calculated as the time from the initiation of the countermovement until the end of the propulsion phase. Jump height was calculated using the vertical velocity of the system center of mass at the instant of take-off (identified as when force <25 N for 300 ms) and the equations of uniformly accelerated motion during the countermovement phase. For ratio metrics, mRSI was calculated as jump height divided by time to take-off, whereas reactive strength index (RSI) was calculated as jump height divided by ground contact time during the CMJ-R. CMJ and CMJ-R metrics were categorized under the PODS acronym described by Ripley et al. [48]: person, outcome, driver, and strategy (Table 1).

For the IMTP, an onset threshold of three SDs of the average force during the one-second weighing period was used to identify the start of the IMTP, as per the manufacturer’s proprietary software settings. The highest vertical force value achieved was recorded as gross peak force, with net peak force calculated by subtracting the system weight taken during the CMJ from the IMTP gross peak force value. Relative net peak force was ratio scaled to body mass (i.e., net peak force/body mass).

### 2.5. Statistical Analyses

Absolute within-session reliability was calculated using the coefficient of variation (CV = SDpooled/group mean × 100) plus 95% confidence intervals (95CI) and interpreted based on the upper bound 95CI (+95CI) as ≤5.00 = excellent, 5.01–10.00 = good, 10.01–15.00 = moderate and >15.00 = poor. Relative within-session reliability was calculated using the intraclass correlation coefficients (ICC, 3,1*k*) plus 95CI and interpreted based on the lower bound 95CI (−95CI) as ≥0.90 = excellent, 0.750–0.899 = good, 0.500–0.749 = moderate, and <0.50 = poor [53].

Normal distribution of the data was determined using Shapiro–Wilk’s test (*p* > 0.05) and presented as mean ± SD. A series of Bayesian independent *t*-tests were performed to compare stronger vs. weaker players using JASP (0.19.3, JASP Team [Computer software]). As there is currently no known threshold to define ‘strong’ from the IMTP in elite female youth footballers, players were split into two groups based on quartiles to ensure a clear difference between groups. The top quartile (*i.e., stronger* = 28.37 ± 3.35 N/kg, *n* = 19) was compared to the bottom quartile (*i.e., weaker* = 15.40 ± 1.45 N/kg, *n* = 23) using relative net peak force from IMTP, with very large differences observed (*g* = 5.05). Relative net peak force was chosen as no differences have been observed across age groups [25].

Directional *t*-tests using a half Cauchy distribution hypothesized that the stronger group would have improved CMJ and CMJ-R performance relating to outcome, driver, and strategy metrics. A default prior was set at 0.707, and robustness regions were reported to determine whether any interpretations from the Bayes factors (BF_10_) were sensitive to the choice of prior distribution. Bayes factors (BF_10_) were interpreted as follows: BF10 = >30.00 = very strong, 10.01–30.00 = strong, 3.01–10.00 = moderate, and 1.00–3.00 = weak evidence supporting the hypothesis (*h*_1_). Evidence to support the null hypothesis (*h*_0_) was interpreted using the following thresholds: BF_10_ = 0.09–0.03 = strong, 0.33–0.10 = moderate, and 1.00–0.33 = weak [54]. To assist in the interpretation of the Bayes analysis, Hedge’s *g* effect sizes and 95% confidence intervals (95%CIs) were calculated and interpreted based on the recommendations of Hopkins [55]: 0.00–0.19 = trivial; 0.20–0.59 = small; 0.60–1.19 = moderate; and ≥1.20 = large.

## 3. Results

Good absolute and relative reliability was observed for IMTP relative net peak force (CV +95CI = 7.03, ICC −95CI = 0.873). Good to excellent absolute and relative reliability were observed for body mass, jump momentum, relative average propulsive force, and propulsive phase duration for all jump variants (CMJ). Moderate to good absolute and relative reliability was observed for all other metrics (Table 2).

The Bayesian independent *t*-tests revealed moderate evidence supporting the null hypothesis (*h*_0_) for CMJ average braking and propulsive force, CMJ force at minimum displacement, and rebound ground contact time, which were trivial to moderate in magnitude (Table 3). Weak evidence was also observed for rebound force at minimum displacement, which was small in magnitude. Very strong evidence supporting the hypothesis (*h*_1_) was observed, with the stronger group demonstrating greater CMJ countermovement depth (Figure 1) and CMJ height during the CMJ portion of the CMJ-R, which were large in magnitude. Strong evidence was also observed with the stronger group achieving greater CMJ time to take-off and rebound jump height (Figure 2) than the weaker group, which were moderate in magnitude (Table 3). All other evidence supporting the hypothesis (*h*_1_) was moderate including RSI (Figure 3) except for body mass (Figure 4), CMJ height, CMJ momentum, rebound average propulsive force, and CMJ braking phase duration (Table 3).

## 4. Discussion

The purpose of this study was to determine if relative maximal isometric force production, measured using the IMTP, can discriminate between higher- and lower-performing jumpers in the CMJ and CMJ-R among female youth soccer players, including phase-specific metrics to better understand the mechanisms responsible for different jump performances, if any. The main findings are that stronger players were heavier (*g* = 0.43) and taller (*g* = 0.94) than weaker players and achieved similar CMJ height (0.27 ± 0.05 m, = 0.25 ± 0.05 m, *g* = 0.45, respectively) by using a greater countermovement depth (0.26 ± 0.05 m, 0.20 ± 0.06 m, *g* = 1.04, respectively) over a longer duration (663.68 ± 89.83 ms, 560.87 ± 128.41 ms, *g* = 0.89). In addition, stronger players achieved greater rebound jump height than weaker players with greater rebound average relative braking force (*g* = 0.82) and similar ground contact times (*g* = 0.17). The higher braking forces are a product of greater jump height during the preceding CMJ leading to a higher velocity on landing, in conjunction with a higher body mass, both resulting in a greater momentum. This information can be useful for practitioners to understand the importance of relative strength and the mechanisms responsible for CMJ and CMJ-R performance, enabling the prescription of appropriate training programs. Despite these differences, it should be acknowledged that maximal isometric peak force was much lower than those reported in male semi-professional and professional English [48] and male Spanish footballers [40], and an important observation is that the players involved in this study could all still be classified as weak (stronger = 28.37 ± 3.35 N/kg, weaker = 15.40 ± 1.45 N/kg). Therefore, the groups herein are considered as ‘stronger’ and ‘weaker’. This further highlights the importance of developing relative strength [24,40], as female footballers can better cope with the increasing physical match demands with increasing age [7] and competition [7,8].

Stronger players were heavier than weaker players, although this difference was small (*g* = 0.49). These findings are in contrast to Ripley et al. [48], who reported moderate differences with stronger players, who were lighter than weaker players. Despite this, the players in the present study are young female footballers, many of whom will be pre-peak height velocity (age range 10.0–18.5 years). Increases in body mass have been reported with increasing age in female youth footballers because of growth and maturation [25,42,43,44]; however, no change in relative strength has been observed despite increases in body mass, jump height, and sprint performance, and therefore momentum [25]. Greater body mass may likely have implications for jump performance, as heavier players will need to apply greater force relative to body mass to achieve the same or greater acceleration than lighter players [56]. Practitioners are therefore encouraged to design training programs aimed to develop relative strength especially during adolescence, given its importance in peak and rapid force production [24,28]. Barr et al. [57] reported small differences in jump height using greater drop heights in stronger compared to weaker female rugby union players (strong = 1RM relative to body mass > 1.00); however these would not usually be considered as ‘strong’ [29,58,59].

There was weak evidence to support the hypothesis that relative strength could distinguish between CMJ height with moderate differences in magnitude (*g* = 0.68). Ripley et al. [48] also reported weak evidence for CMJ height in professional and semi-professional male soccer players. CMJ height is the most commonly reported metric used by practitioners in team sports; however based on the results of the present study and those reported by Ripley et al. [48], the limitations of only reporting jump height are highlighted, as this reflects a global measure of jump outcome rather than performance. Interestingly, strong evidence was observed to support the importance of relative strength on rebound jump height (BF_10_ = 15.843, *g* = 0.86). This could be explained by very strong evidence supporting the hypothesis that stronger players achieved greater jump height during the countermovement portion of the CMJ-R (*g* = 1.17), hence falling from a greater height, which, based on the impulse momentum relationship, increases the demands of the rebound portion of the jump. These findings are similar to Barr et al. [57], who reported a difference in fall height between stronger and weaker female rugby union players. Stronger players demonstrate superior explosive lower limb movements due to enhanced neuromuscular and structural adaptations [24,26,58,60,61]; it is plausible that stronger players in the present study were able to achieve greater rebound jump height due to these adaptations despite being considered weak by the previous researchers [40,48]. Jump momentum (i.e., take-off velocity x body mass) was weak to moderate evidence to support the hypothesis for the CMJ (*g* = 0.72) and CMJ-R (*g* = 0.77), which were moderate in magnitude, respectively, which is most likely explained by differences in body mass between the two groups. Moderate evidence was observed for mRSI supporting the null hypothesis, whereas RSI had moderate evidence supporting the hypothesis. Small differences were observed for the mRSI, with weaker players achieving greater mRSI outputs than stronger players, whereas moderate differences in RSI were observed, with superior scores in stronger players. These findings present a potential limitation to the mRSI, with weaker players achieving superior scores than stronger players despite lower CMJ height, thus adopting a different strategy [62]. Krzyszkowski et al. [45] assessed phase-specific predictors of CMJ performance in good and bad jumpers, based on mRSI in male collegiate basketballers. The high mRSI group (i.e., good jumpers) had greater jump height with shorter time to take-off, whereas our findings are influenced by time to take-off more than jump height, which was moderate in magnitude between stronger and weaker players (*g* = 0.86). Practitioners are reminded to consider the constituent parts of ratio metrics to better understand the mechanisms responsible for CMJ performance. Moderate differences were observed for RSI (*g* = 0.78), which supports the importance of relative strength in CMJ-R jump performance. Greater RSI in stronger players, who were also heavier, is likely explained by the greater jump heights with similar ground contact times. Female youth footballers should aim to develop relative strength in order to enhance plyometric ability, which relates to goal-scoring opportunities [16] and to cope with the increasing physical demands of football with age [7] and competition [7,8].

There was moderate evidence to support the null hypothesis for CMJ diver metrics (average braking and propulsive force and force at minimum displacement), which were small to moderate in magnitude (*g* = 0.48–0.86). Ripley et al. [48] reported moderate evidence to support the hypothesis for relative isometric strength when discriminating between CMJ average braking forces (BF_10_ = 6.017, *d* = 0.56). Further, both stronger and weaker players reported by Ripley et al. [48] exhibited greater CMJ average braking and propulsive force than those in the present study, except for the strong players CMJ average propulsive force (19.37 ± 2.65 N/kg vs. 19.80 ± 3.35 N/kg, respectively). Our CMJ findings are greater than those reported in trained/developmental girl’s youth soccer players [63] and female collegiate soccer players [64], which may be reflective of the superior performance level of subjects in the current study and greater training exposure to strength and conditioning in comparison to Bright et al. [63] and Thomas et al. [64]. IMTP values of the strong group in the present study were greater than those reported by Thomas et al. [64] but lower than Ripley et al. [48] (23.49 ± 5.30 N/kg and 33.41 ± 4.93 N/kg, respectively). Moderate evidence supporting the hypothesis was observed for rebound average braking force with moderate differences in magnitude (*g* = 0.82). However, weak evidence was observed for rebound force at minimum displacement (*h*_0_) and rebound average propulsive force (*h*_1_) with a small difference in magnitude (*g* = 0.28–0.52). These findings may be explained by the large differences in fall height from the CMJ portion of the CMJ-R, meaning that stronger players will have been required to exhibit greater average braking force during the rebound task to decelerate a greater body mass from a greater height. This greater fall momentum yet greater rebound jump height further highlights the superior stretch–shortening cycle function in stronger players, which is utilized better in females compared to males [37]. Female youth footballers should aim to develop relative strength to enhance stretch–shortening cycle ability, with adaptations possible in weaker players through appropriate strength training [58,59].

Strong to very strong evidence supporting the hypothesis was observed for time to take-off (*g* = 0.89) and countermovement depth (*g* = 1.04), respectively. This may explain why driver metrics (i.e., relative average braking and propulsive force) were reduced in stronger players due to adopting a compliant strategy to achieve a similar relative propulsive net impulse (force × time). Our findings contrast Ripley et al. [48], who reported weak evidence for time to take-off (*h*_1_) and countermovement depth (*h*_0_), whereby stronger players executed the CMJ with shorter time to take-off and countermovement depth. Stronger players in the present study exhibited a greater countermovement depth, and therefore time to take-off, to achieve similar jump heights than the weaker players. This may be explained by the greater body mass of stronger players, where the increased countermovement depth would allow for force to be applied over a longer duration to achieve the appropriate acceleration and jump height [56,62]. Countermovement depth and time to take-off in the present study was lower than those reported by Ripley et al. [48] and Thomas [64], which may be the result of different maturity statuses and statures between the studies; for example, a 20 cm countermovement depth for a less mature player will lead to a greater relative countermovement depth (i.e., countermovement depth/standing stature) than that in a fully developed player. However, this was not examined in this study but may be worth considering by future researchers. There was moderate evidence supporting the null hypothesis for rebound ground contact time, which when taken in context, demonstrated that stronger players exhibited shorter ground contact times than weaker players (*g* = 0.17). Moderate evidence was also observed for rebound depth (*h*_1_). Stronger players were able to achieve greater rebound jump heights by decelerating and reaccelerating a heavier mass from a greater fall height with shorter ground contact times. The results of the present study highlight the importance of relative strength on stretch–shortening cycle function [24,26,58,61]. Further, as football incurs time pressure situations, rebound tasks are crucial for important match events (i.e., goal scoring opportunities) which are often associated [16]. Therefore, female footballers should develop relative maximal strength to improve stretch–shortening cycle function and sporting performance [24,29].

A limitation of the present study is that only top and bottom quartiles were used to distinguish ‘*stronger*’ and ‘*weaker*’ players, meaning the middle quartiles were excluded from the analysis. The rationale to include top and bottom only quartiles was to create two clear groups, as youth players will train and compete against each other, and there are currently no known thresholds to define ‘strong’ in elite female youth footballers using the IMTP. Future researchers should aim to develop benchmarks for peak and rapid force for elite female youth footballers using the IMTP so that exclusion from analysis is avoided. A second limitation is that data is collected from a single club, which may not be representative of other girls’ youth soccer academies. Further, there is currently no known threshold to define ‘strong’ using the IMTP in elite female youth footballers. Future researchers should aim to collaborate and develop multi-club studies to better capture the importance of relative strength on jump performance in female youth footballers. A third limitation is that age and maturity status were not controlled for; however, previous researchers have demonstrated no changes in relative peak force using IMTP with increasing age and maturation in elite female youth footballers [25,42]. Future work should consider controlling for age and maturation status to offer more insights into the importance of relative strength on CMJ and MCJ-R performance during adolescence. Another limitation is that the results are representative of a single time point in the season. Although mid-season was chosen to allow players to accumulate training and competitive match loads, this may not reflect best practice. Further research is needed to monitor how changes in relative strength during a season impact phase-specific jump performance in the CMJ and CMJ-R. Also, as the rebound portion of the jump is an ankle-dominant task, evaluation of the difference in maximal and rapid plantar flexion force production and its association with rebound jump performance would be useful in the future. A final limitation of the present study is that the players could be categorized as weak in comparison to professional and semi-professional English male footballers [48] and professional Spanish male footballers [40]. Despite this, players are categorized as stronger and weaker based on a very large difference in magnitude (*g* = 5.05).

## 5. Conclusions

Maximal relative isometric strength did not have implications on CMJ outcome (i.e., jump height) but was able to discriminate CMJ-R performance. Stronger players, who were heavier, achieved similar jump heights to weaker players by adopting a more compliant strategy (i.e., greater countermovement depth and time to take-off) and applying force over a longer duration to achieve a greater relative net impulse and therefore resulting velocity. Weaker players demonstrated greater mRSI, a measure of jump efficiency, which provided weak evidence supporting the hypothesis (*h*_1_) and presents a limitation of mRSI, with practitioners reminded to consider the constituent parts to better understand the mechanisms responsible for CMJ performance. Maximal relative isometric strength has implications on CMJ-R jump outcome (i.e., rebound jump height). Strong to very strong evidence supporting the hypothesis (*h*_1_) with moderate to large magnitudes was demonstrated for fall height and rebound jump height in stronger players with similar ground contact times. This further highlights the importance of developing relative strength, as rapid force production is required in football for acceleration, deceleration, and changes in direction. Contextually, in a goal-scoring scenario, if a stronger player can rapidly apply force during a braking (i.e., deceleration) action and change direction, this could result in favorable match events (i.e., goal-scoring opportunities). Therefore, female youth soccer players should increase relative isometric strength to enhance fast stretch–shortening cycle function (i.e., ≤250 ms). Practitioners, however, should be cautious when interpreting CMJ and CMJ-R PODS metrics in isolation, considering relative strength alongside these (i.e., are players strong enough?).

## Figures and Tables

**Figure 1 muscles-04-00023-f001:**
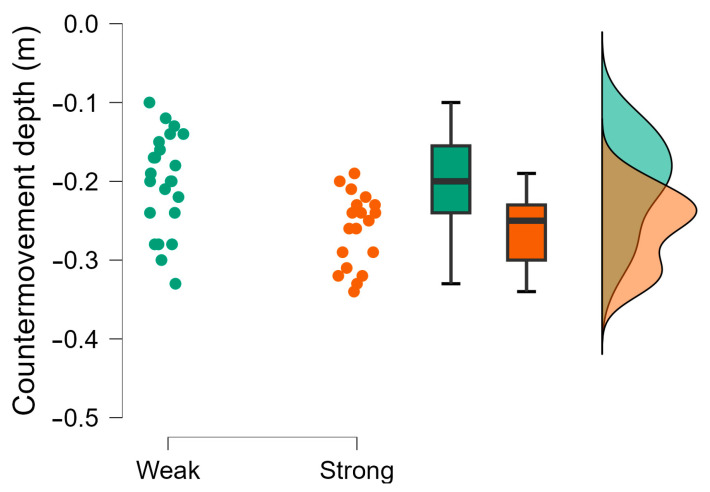
CMJ countermovement depth (m) raincloud plot with individual data points, box and whisker plots presenting median, interquartile range, and normal distribution for weaker and stronger groups. Green dots refer to individual data points for weaker players; Orange dots refer to individual data points for stronger players.

**Figure 2 muscles-04-00023-f002:**
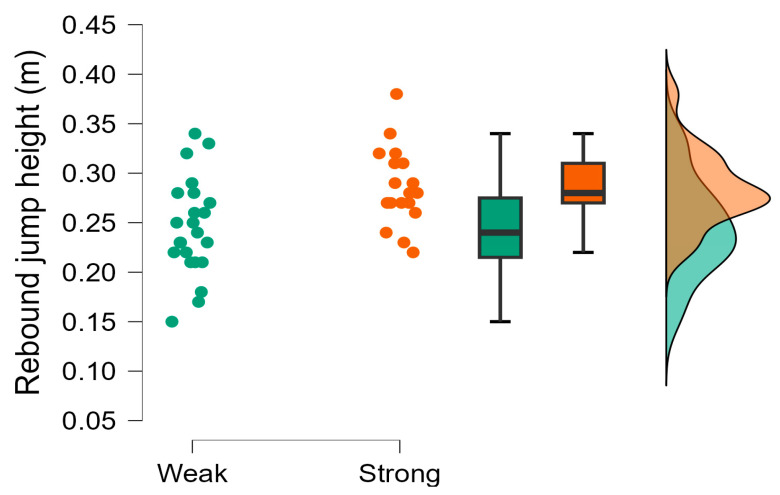
Rebound jump height (m) raincloud plot with individual data points, box and whisker plots presenting median, interquartile range, and normal distribution for weaker and stronger groups.

**Figure 3 muscles-04-00023-f003:**
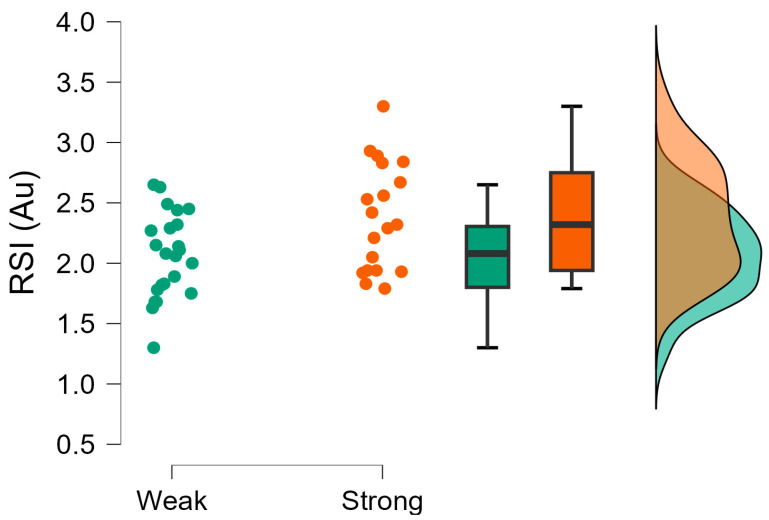
Reactive strength index (RSI, Au) raincloud plot with individual data points, box and whisker plots presenting median, interquartile range, and normal distribution for weaker and stronger groups.

**Figure 4 muscles-04-00023-f004:**
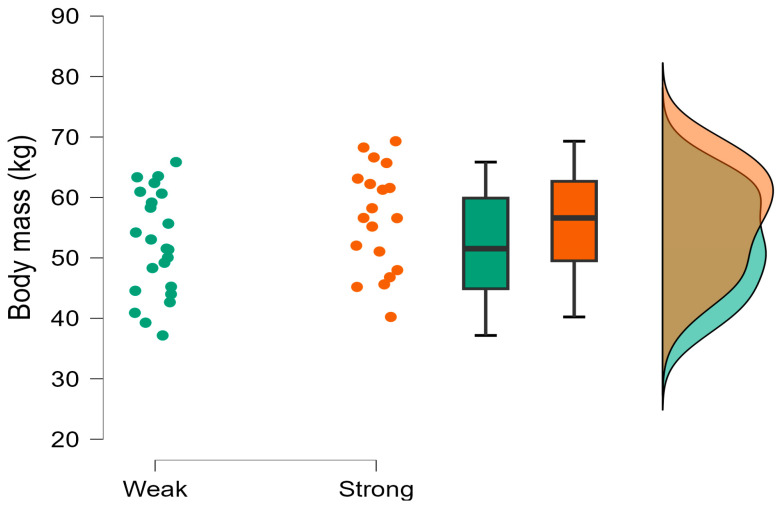
Body mass (kg) raincloud plot with individual data points, box and whisker plots presenting median, interquartile range, and normal distribution for weaker and stronger groups. Green dots refer to individual data points for weaker players; Orange dots refer to individual data points for stronger players.

**Table 1 muscles-04-00023-t001:** Countermovement jump and countermovement rebound jump metrics selected for Bayesian analysis using PODS acronym [48].

Category	Metric
Person	Body mass (kg)
Outcome	Jump height (m) Rebound jump height (m) Jump momentum (kg.m/s^−1^) Rebound jump momentum (kg.m/s^−1^) mRSI (Au) RSI (Au)
Driver	Average braking force (N/kg) Rebound average braking force (N/kg) Average propulsive force (N/kg) Rebound average propulsive force (N/kg) Force at minimum displacement (N/kg) Rebound force at minimum displacement (N/kg)
Strategy	Braking phase duration (s) Propulsive phase duration (s) Time to take-off (s) Ground contact time (s) Countermovement depth (m) Rebound depth (m)
mRSI = modified reactive strength index, RSI = reactive strength index	

**Table 2 muscles-04-00023-t002:** CMJ and CMJ-R absolute and relative reliability.

Metric	Jump Variant	ICC (95CI)	CV (95CI)
**Person Metric**			
Body mass (kg)	CMJ	1.000 (1.000, 1.000)	0.25 (0.23, 0.27)
**Outcome Metrics**			
Jump height (m)	CMJ	0.927 (0.898, 0.927)	3.89 (3.60, 4.18)
	CMJ-R CMJ	0.901 (0.863, 0.930)	5.12 (4.74, 5.50)
	CMJ-RJ	0.756 (0.675, 0.823)	7.15 (6.61, 7.68)
Jump momentum (kg·m/s^−1^)	CMJ	0.992 (0.988, 0.992)	1.77 (1.64, 1.90)
	CMJ-RJ	0.945 (0.924, 0.962)	3.64 (3.37, 3.91)
mRSI (Au) RSI	CMJ	0.808 (0.742, 0.808)	8.93 (8.270, 9.60)
	CMJ-RJ	0.747 (0.664, 0.816)	9.19 (8.50, 9.87)
**Driver Metrics**			
Average braking force (N/kg)	CMJ	0.792 (0.721, 0.850)	5.56 (5.15, 5.98)
	CMJ-RJ	0.713 (0.623, 0.790)	7.29 (6.75, 7.84)
Average propulsive force (N/kg)	CMJ	0.889 (0.848, 0.922)	3.37 (3.12, 3.62)
	CMJ-RJ	0.703 (0.610, 0.782)	6.26 (5.80, 6.73)
Force at minimum displacement (N/kg)	CMJ	0.852 (0.798, 0.895)	5.78 (5.35, 6.21)
	CMJ-RJ	0.703 (0.611, 0.782)	9.53 (8.82, 10.24)
**Strategy Metrics**			
Braking phase (s)	CMJ	0.669 (0.606, 0.779)	10.70 (9.90, 11.50)
Propulsive phase (s)	CMJ	0.882 (0.838, 0.917)	6.19 (5.73, 6.65)
Time to take-off (ms) Rebound ground contact time (ms)	CMJ	0.755 (0.741, 0.862)	7.69 (7.12, 8.26)
	CMJ-RJ	0.715 (0.625, 0.791)	9.43 (8.73, 10.13)
Countermovement depth (m)	CMJ	0.883 (0.839, 0.917)	7.92 (7.330, 8.51)
	CMJ-RJ	0.484 (0.360, 0.602)	−32.91 (−30.46, −35.36)

ICC = intraclass correlation coefficient, CV = coefficient of variation, 95CI = 95% confidence intervals, mRSI = modified reactive strength index, RSI = reactive strength index, CMJ = countermovement jump, CMJ-RJ = rebound portion of the countermovement rebound jump. Green = excellent, light green = good, amber = moderate, red = poor.

**Table 3 muscles-04-00023-t003:** Mean ± SD, Bayesian independent *t*-test (BF10) and Cohen’s d effect size difference for “strong” vs. “weak” players.

Metric	Mean ± SD	BF_10_	Bayes Interpretation	Hedges *g* (95CI)
Body Mass (kg)	**Stronger** = 56.51 ± 8.65, **Weaker** = 52.23 ± 8.55	1.542	Weak evidence *h*_1_	0.50 (0.47, 0.54)
CMJ height (m)	**Stronger** = 0.27 ± 0.05, **Weaker** = 0.25 ± 0.05	1.204	Weak evidence *h*_1_	0.68 (0.64, 0.72)
Rebound CMJ (m)	**Stronger** = 0.27 ± 0.04, **Weaker** = 0.22 ± 0.05	106.265	Very Strong evidence *h*_1_	1.17 (1.11, 1.23)
Rebound jump height (m)	**Stronger** = 0.29 ± 0.04, **Weaker** = 0.25 ± 0.05	15.843	Strong evidence *h*_1_	0.86 (0.82, 0.91)
CMJ momentum (kg.m/s^−1^)	**Stronger** = 129.63 ± 23.52, **Weaker** = 115.41 ± 24.39	2.415	Weak evidence *h*_1_	0.72 (0.68, 0.76)
Rebound jump momentum (kg.m/s^−1^)	**Stronger** = 133.69 ± 23.66, **Weaker** = 114.44 ± 25.37	6.890	Moderate evidence *h*_1_	0.77 (0.72, 0.81)
mRSI (Au)	**Stronger** = 0.41 ± 0.07, **Weaker** = 0.47 ± 0.14	0.130	Moderate evidence *h*_0_	−0.17 (−0.20, −0.14)
RSI (Au)	**Stronger** = 2.38 ± 0.45, **Weaker** = 2.06 ± 0.35	7.580	Moderate evidence *h*_1_	0.78 (0.74, 0.82)
CMJ average braking force (N/kg)	**Stronger** = 18.63 ± 2.02, **Weaker** = 19.80 ± 3.35	0.148	Moderate evidence *h*_0_	−0.48 (−0.52, −0.47)
Rebound average braking force (N/kg)	**Stronger** = 35.51 ± 5.42, **Weaker** = 31.82 ± 3.39	9.589	Moderate evidence *h*_1_	0.82 (0.77, 0.86)
CMJ average propulsive force (N/kg)	**Stronger** = 19.28 ± 1.16, **Weaker** = 20.97 ± 2.99	0.106	Moderate evidence *h*_0_	−0.51 (−0.54, −0.47)
Rebound average propulsive force (N/kg)	**Stronger** = 30.59 ± 3.31, **Weaker** = 28.78 ± 3.51	1.785	Weak evidence *h*_1_	0.52 (0.48, 0.55)
CMJ force at minimum displacement (N/kg)	**Stronger** = 23.53 ± 2.31, **Weaker** = 26.89 ± 5.36	0.100	Moderate evidence *h*_0_	−0.86 (−0.91, −0.81)
Rebound force at minimum displacement (N/kg)	**Stronger** = 51.55 ± 8.82, **Weaker** = 49.22 ± 7.59	0.667	Weak evidence *h*_0_	0.28 (0.25, 0.31)
CMJ braking phase duration (ms)	**Stronger** = 144.74 ± 20.38, **Weaker** = 123.04 ± 45.27	2.491	Weak evidence *h*_1_	0.78 (0.74, 0.83)
CMJ propulsive phase duration (ms)	**Stronger** = 244.74 ± 27.96, **Weaker** = 210.43 ± 49.03	9.778	Moderate evidence *h*_1_	0.84 (0.80, 0.89)
CMJ time to take-off (ms)	**Stronger** = 663.68 ± 89.83, **Weaker** = 560.87 ± 128.41	15.902	Strong evidence *h*_1_	0.84 (0.80, 0.89)
Rebound ground contact time (ms)	**Stronger** = 213.90 ± 34.98, **Weaker** = 219.36 ± 26.57	0.212	Moderate evidence *h*_0_	−0.17 (−0.20, −0.15)
CMJ countermovement depth (m)	**Stronger** = 0.26 ± 0.05, **Weaker** = 0.20 ± 0.06	55.734	Very strong evidence *h*_1_	−1.15 (−1.11, −1.00)
Rebound depth (m)	**Stronger** = 0.09 ± 0.03, **Weaker** = 0.08 ± 0.02	0.142	Moderate evidence *h*_1_	−0.43 (−0.47, −0.40)

95CI = 95% confidence intervals; BF_10_ = Bayes factor; *h*_1_ = supporting the hypothesis; *h*_0_ = supporting the null hypothesis.

## Data Availability

Data are available on reasonable request to the corresponding author.

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
