# Peer review of "Lower Limb Muscle Strength Matters: Effect of Relative Isometric Strength on Countermovement and Rebound Jump Performance in Elite Youth Female Soccer Players"

_muscles, 2025, doi:10.3390/muscles4030023_

Round 1

Reviewer 1 Report

Comments and Suggestions for Authors

The article titled "Strength Matters: Effect of Relative Isometric Strength on Countermovement Jump Performance in Elite Youth Female Soccer Players" is a well-conceived and conducted investigation. The introduction and discussion are comprehensive, and the methods are solid.

There is one major concern regarding the full clarity of the purpose statement in the introduction section, and some minor points. By addressing these few points, this article can further improve and be a valid investigation in the context of sports science research.

Major comments:

Although briefly reported, the aim of the present study is explicitly stated only in the initial part of the Discussion section. Since the purpose statement and the hypotheses shall pose the basis for your research, which is well-conducted, a better understanding of the purpose would reinforce the final part of your introduction.

Minor comments:

Abstract:

Typo: Expression is repeated twice: "expression of maximal and rapid force expression"

The conclusion can be smoother by stating that  "Rebound jump performance benefits from the development of rapid force production relative to the available force capacity " (or relative to the max force)

Introduction:

Line 66: In this context, a recent article provided important information regarding the development of maximal and rapid force production in both male and female athletes (although not done in football). It would be relevant to cite and consider it for the following part of the introduction and discussion (https://doi.org/10.1007/s00421-024-05604-1).

Line 72: Incomplete sentence after  "resulting velocity and... "

Line 77: typo  "Netwon’s third law "

Methods:

Line 224: SD has already been introduced above as standard deviation, in line 191

Author Response

Reviewer 1

Comments and Suggestions for Authors

The article titled "Strength Matters: Effect of Relative Isometric Strength on Countermovement Jump Performance in Elite Youth Female Soccer Players" is a well-conceived and conducted investigation. The introduction and discussion are comprehensive, and the methods are solid.

There is one major concern regarding the full clarity of the purpose statement in the introduction section, and some minor points. By addressing these few points, this article can further improve and be a valid investigation in the context of sports science research.

Response: Thank you for your time, consideration and feedback on our manuscript. We have considered all comments and have amended changes relating to your feedback with yellow highlighted text in the updated manuscript.

We have extended the purpose statement in the introduction. Thank you for this suggestion.

Major comments:

Although briefly reported, the aim of the present study is explicitly stated only in the initial part of the Discussion section. Since the purpose statement and the hypotheses shall pose the basis for your research, which is well-conducted, a better understanding of the purpose would reinforce the final part of your introduction.

Response: Thank you for this comment. We have extended the purpose statement of the introduction to include: “Therefore, the purpose of this study was to determine the effect of relative strength on CMJ and CMJ-R performance in elite female youth footballers, to offer practitioners valuable insights on the importance of relative strength on CMJ and CMJ-R performance.”

Minor comments:

Abstract:

Comment: Typo: Expression is repeated twice: "expression of maximal and rapid force expression"

Response:  Thank you for this comment and observation. We have amended the manuscript to the following: "expression of maximal and rapid force are important for athletic performance in female soccer”

Comment: The conclusion can be smoother by stating that  "Rebound jump performance benefits from the development of rapid force production relative to the available force capacity " (or relative to the max force)

 Response: Thank you for this suggestion. We agree that your suggestion is a viable option, unfortunately its inclusion over the original statement exceeds to journal abstract word limit guidelines, and the information prior to this is important for the reader to understand the study purpose, methods and key findings. We have decided to keep the conclusion statement as the original statement but express our gratitude towards your suggestion.

Introduction:

Comment:  Line 66: In this context, a recent article provided important information regarding the development of maximal and rapid force production in both male and female athletes (although not done in football). It would be relevant to cite and consider it for the following part of the introduction and discussion (https://doi.org/10.1007/s00421-024-05604-1).

Response: Thank you for this suggestion. We have included this citation in the introduction and discussion to support the importance of developing rapid force production.

Comment:  Line 72: Incomplete sentence after  "resulting velocity and... "

Response: We have amended the manuscript, removing the word ‘and,’ to complete the sentence

Comment : Line 77: typo  "Netwon’s third law "

Response: Amended to include the apostrophe.

Methods:

Comment: Line 224: SD has already been introduced above as standard deviation, in line 191

Response: Amended to mean ± SD

Reviewer 2 Report

Comments and Suggestions for Authors

Peer Review

Article
"Strength Matters: Effect of Relative Isometric Strength on Countermovement Jump Performance in Elite Youth Female Soccer Players"

Dear authors,

It is noteworthy to acknowledge the significance of your work in this area, which is a crucial addition to the existing body of literature on youth female athletic development. The analysis of relative isometric strength and jump phase performance metrics is timely and methodologically promising. However, a significant omission related to participant analysis was identified, and your careful attention is required.

In the following review, I offer constructive recommendations, particularly regarding participant selection, statistical scope, and interpretation, with the aim of enhancing clarity, transparency, and impact.

Introduction

Comment 1: The title of the article suggests that the study focuses on "elite youth female soccer players" and links relative strength to CMJ performance. However, the introduction does not clearly explain why only the extreme quartiles (top and bottom 25%) were used for analysis later. The exclusion of this rationale gives rise to an inconsistency between the sample size (n = 96) and the final analytical sample (n = 42).

Comment 2: The hypothesis is stated in general terms ("stronger players would achieve superior CMJ performance"), but it does not explicitly anticipate differences in strategy metrics (e.g., countermovement depth or time to take-off). However, these metrics are subsequently identified as significant factors in the conclusions.

Materials and Methods

Participants & Sampling

Comment 3: The initial sample comprised 96 participants; however, only 42 were subjected to analysis. The remaining 54 participants (middle two quartiles) were excluded without justification, which introduces selection bias. However, there is a conspicuous absence of any elaboration on the potential implications of this exclusion on the validity, representativeness, or variance of the data. This omission instigates apprehensions regarding transparency and generalizability.

Jump and Strength Testing

Comment 4: The methodology does not elucidate how jump technique (e.g., depth, posture, hand placement) was standardized beyond verbal instruction and trial repetition. Additionally, and equally significant, there is an absence of explicit control or reporting of maturity status. However, the age range encompasses both pre- and post-peak height velocity. This critical omission is particularly salient in the context of the adolescent female population.

CMJ-R Protocol.

Comment 5: Despite the CMJ-R's utilization of individualized fall heights derived from preceding CMJs, the manuscript fails to elucidate the methodology employed to ensure consistency across trials. Additionally, it does not address whether variability in CMJ heights influences the interpretation of rebound. A thorough elucidation of this phenomenon is requisite.

Statistical Analyses

Comment 6: The decision to utilize solely the top and bottom quartiles should have been explicitly substantiated as a trade-off between contrast and sample representativeness.

Comment 7: One-tailed Bayesian t-tests are utilized extensively; however, the decision to assume directional effects for all outcome variables is not adequately supported.

Results

Comment 8: In the statistical analyses section, the terms "players" and "groups" are used repeatedly, as if the entire sample of 96 subjects had been analyzed, when in fact only 42 subjects were included in the study. This may mislead readers.

Comment 9: A visual breakdown (e.g., CONSORT-style flow diagram) or participant distribution table is missing. This would help clarify how the final analytical sample was determined and how excluded participants differed, if at all.

Discussion

Comment 10: The exclusion of more than half of the sample is not addressed, nor is its impact on the robustness of the findings discussed. This represents a significant constraint that merits transparent acknowledgement.

Comment 11: However, disparities in maturity, age, and body composition among the groups are not statistically controlled or discussed, despite their probable influence on performance and strength characteristics.

Comment 12: While the CMJ-R outcomes are given significant attention, the discussion fails to adequately address the reasons why relative strength did not differentiate CMJ height or driver metrics (e.g., propulsive force).

Conclusions

Comment 13: Recommendations for practitioners imply broader generalizability than the data supports.

General recommendations for the authors:
 I recommend that the authors explicitly:

  • Justify the rationale for excluding the middle 50% of participants;
  • Discuss the potential statistical and practical implications of this exclusion;
  • Include sensitivity analyses (e.g., full-cohort regression) if available, or acknowledge this as a key limitation.

Without these clarifications, the actual analytical sample is 42—not 96 as presented—and this could be misleading to readers and reviewers.

Author Response

Reviewer 2

Dear authors,

It is noteworthy to acknowledge the significance of your work in this area, which is a crucial addition to the existing body of literature on youth female athletic development. The analysis of relative isometric strength and jump phase performance metrics is timely and methodologically promising. However, a significant omission related to participant analysis was identified, and your careful attention is required.

In the following review, I offer constructive recommendations, particularly regarding participant selection, statistical scope, and interpretation, with the aim of enhancing clarity, transparency, and impact.

Response: Thank you for your time, consideration and feedback on our manuscript. We have considered all comments and have amended changes relating to your feedback with green highlighted text in the updated manuscript. We have provided an itemised response to your comments, including the comment surrounding participant omission. In short, we wished to investigate the effect of relative strength and jump and rebound performance with no known threshold for strong in this population, we selected top and bottom quartiles to distinguish two clear groups. This is emphasised in the manuscript and identified as a limitation in the discussion.

Introduction

Comment 1: The title of the article suggests that the study focuses on "elite youth female soccer players" and links relative strength to CMJ performance. However, the introduction does not clearly explain why only the extreme quartiles (top and bottom 25%) were used for analysis later. The exclusion of this rationale gives rise to an inconsistency between the sample size (n = 96) and the final analytical sample (n = 42).

Response: Thank you for this comment and we agree.  As we wished to investigate the effect of relative strength and jump and rebound performance with no known threshold for strong in this population, we selected top and bottom quartiles to distinguish two clear groups. This is emphasised in the manuscript and identified as a limitation in the discussion.

Comment 2: The hypothesis is stated in general terms ("stronger players would achieve superior CMJ performance"), but it does not explicitly anticipate differences in strategy metrics (e.g., countermovement depth or time to take-off). However, these metrics are subsequently identified as significant factors in the conclusions.

Response: Thank you for this comment and we agree. The manuscript has been amended to extend the hypothesis beyond jump performance.   

Materials and Methods

Participants & Sampling

Comment 3: The initial sample comprised 96 participants; however, only 42 were subjected to analysis. The remaining 54 participants (middle two quartiles) were excluded without justification, which introduces selection bias. However, there is a conspicuous absence of any elaboration on the potential implications of this exclusion on the validity, representativeness, or variance of the data. This omission instigates apprehensions regarding transparency and generalizability.

Response: Thank you for this comment and as per comment 1, we wished to distinguish two clear groups as there is no known threshold for strong in this population. Therefore we selected top and bottom quartiles, however this is acknowledged as a limitation in the discussion.

Jump and Strength Testing

Comment 4: The methodology does not elucidate how jump technique (e.g., depth, posture, hand placement) was standardized beyond verbal instruction and trial repetition. Additionally, and equally significant, there is an absence of explicit control or reporting of maturity status. However, the age range encompasses both pre- and post-peak height velocity. This critical omission is particularly salient in the context of the adolescent female population.

 Response: Thank you for this comment. Regarding standardization of jump technique, we have outlined that CMJ and CMJ-R was performed with arms akimbo (line 170) and have included information relating to self-selected depth. This has been recommended by previous researchers:

  • Petrigna, L., Karsten, B., Marcolin, G., Paoli, A., D’Antona, G., Palma, A., & Bianco, A. (2019). A review of countermovement and squat jump testing methods in the context of public health examination in adolescence: reliability and feasibility of current testing procedures. Frontiers in physiology10, 1384.
  • Bishop, C., Turner, A., Jordan, M., Harry, J., Loturco, I., Lake, J., & Comfort, P. (2022). A framework to guide practitioners for selecting metrics during the countermovement and drop jump tests. Strength & Conditioning Journal44(4), 95-103.

Regarding maturity status, Emmond et al. (2017, 2018) demonstrated that relative strength did not change with increasing age and maturation. Therefore, we did not control for maturation status but have identified this a limitation and recommendation for future research in the discussion.

  • Emmonds, S., Morris, R., Murray, E., Robinson, C., Turner, L., & Jones, B. (2017). The influence of age and maturity status on the maximum and explosive strength characteristics of elite youth female soccer players. Science and Medicine in Football1(3), 209-215.
  • Emmonds, S., Till, K., Redgrave, J., Murray, E., Turner, L., Robinson, C., & Jones, B. (2018). Influence of age on the anthropometric and performance characteristics of high-level youth female soccer players. International Journal of Sports Science & Coaching13(5), 779-786.

CMJ-R Protocol.

Comment 5: Despite the CMJ-R's utilization of individualized fall heights derived from preceding CMJs, the manuscript fails to elucidate the methodology employed to ensure consistency across trials. Additionally, it does not address whether variability in CMJ heights influences the interpretation of rebound. A thorough elucidation of this phenomenon is requisite.

 Response: Thank you for noting this. We have now included the reliability of the CMJ height from the countermovement portion of the CMJ-R with good absolute and relative reliability identified (Table 2). We have added further details in the methods about consistency across trials.

Statistical Analyses

Comment 6: The decision to utilize solely the top and bottom quartiles should have been explicitly substantiated as a trade-off between contrast and sample representativeness.

Response: As per previous responses, we wished to distinguish two clear groups as there is no known threshold for strong in this population. Therefore, we selected top and bottom quartiles, however this is acknowledged as a limitation in the discussion.

Comment 7: One-tailed Bayesian t-tests are utilized extensively; however, the decision to assume directional effects for all outcome variables is not adequately supported.

Response: Thank you for this comment. We decided to use one-tailed Bayesian t-tests as we expected improvements in one direction based on the underpinning biomechanics and the observation from Ripley et al (2025).

  • Ripley, N. J., Fahey, J., Hassim, N., & Comfort, P. (2025). Effect of Relative Isometric Strength on Countermovement Jump Performance in Professional and Semi-Professional Soccer Players. Biomechanics5(2), 32.

Results

Comment 8: In the statistical analyses section, the terms "players" and "groups" are used repeatedly, as if the entire sample of 96 subjects had been analyzed, when in fact only 42 subjects were included in the study. This may mislead readers.

Response: Thank you for this, we have amended this.

Comment 9: A visual breakdown (e.g., CONSORT-style flow diagram) or participant distribution table is missing. This would help clarify how the final analytical sample was determined and how excluded participants differed, if at all.

Response: Thank you for this suggestion. After careful consideration of the journal guidelines, we have decided not to include this, as this is a requirement for randomized clinical trials. However, we thank you for this developmental point and will consider this for future work relating to this design.

Discussion

Comment 10: The exclusion of more than half of the sample is not addressed, nor is its impact on the robustness of the findings discussed. This represents a significant constraint that merits transparent acknowledgement.

Response: As per previous responses, we wished to distinguish two clear groups as there is no known threshold for strong in this population. Therefore, we selected top and bottom quartiles, however this is acknowledged as a limitation in the discussion.

Comment 11: However, disparities in maturity, age, and body composition among the groups are not statistically controlled or discussed, despite their probable influence on performance and strength characteristics.

Response: Thank you for this comment. In the discussion we have outlined differences in body mass and stature (lines 296-7). Regarding maturity status, Emmonds et al. (2017, 2018) demonstrated that relative strength did not change with increasing age and maturation. Therefore we did not control for maturation status but have included this as a limitation and recommendation for future research in the discussion.

Emmonds, S., Morris, R., Murray, E., Robinson, C., Turner, L., & Jones, B. (2017). The influence of age and maturity status on the maximum and explosive strength characteristics of elite youth female soccer players. Science and Medicine in Football1(3), 209-215.

Emmonds, S., Till, K., Redgrave, J., Murray, E., Turner, L., Robinson, C., & Jones, B. (2018). Influence of age on the anthropometric and performance characteristics of high-level youth female soccer players. International Journal of Sports Science & Coaching13(5), 779-786.

Comment 12: While the CMJ-R outcomes are given significant attention, the discussion fails to adequately address the reasons why relative strength did not differentiate CMJ height or driver metrics (e.g., propulsive force).

 Response: Thank you for this comment. We believe that this has been addressed in the manuscript, but have added additional details (line 398):

Strong to very strong evidence supporting the hypothesis was observed for time to take-off (g = 0.89) and countermovement depth (g = 1.04), respectively. This may explain why driver metrics (i.e. relative average braking and propulsive force) were reduced in stronger players due to adopting a compliant strategy to achieve a similar relative propulsive net impulse (force x time).

Conclusions

Comment 13: Recommendations for practitioners imply broader generalizability than the data supports.

Response: Thank for noting this, however we respectfully disagree. The cohort of player would train and compete against each other (and similar players from opposing teams). Therefore our findings and conclusion highlight the importance of developing relative strength in female youth footballers.

General recommendations for the authors:
 I recommend that the authors explicitly:

  • Justify the rationale for excluding the middle 50% of participants;
  • Discuss the potential statistical and practical implications of this exclusion;
  • Include sensitivity analyses (e.g., full-cohort regression) if available, or acknowledge this as a key limitation.

Without these clarifications, the actual analytical sample is 42—not 96 as presented—and this could be misleading to readers and reviewers

Response:  Thank you for this comment and we agree.  As we wished to investigate the effect of relative strength and jump and rebound performance with no known threshold for strong in this population, we selected top and bottom quartiles to distinguish two clear groups. This is emphasised in the manuscript and identified as a limitation in the discussion. With regarding further analysis, we believe that this is not necessary as our findings support the conclusions.